

# EEG-based schizophrenia diagnosis using deep learning with multi-scale and adaptive feature selection

Alanoud Al Mazroa[1], Majdy M. Eltahir[2], Shouki A. Ebad[3], Faiz Abdullah Alotaibi[4], Venkatachalam K[5] and Jaehyuk Cho[6]

[1] Department of Information Systems, Princess Nourah bint Abdulrahman University, Riyadh, Saudi Arabia
[2] Department of Information Systems, King Khalid University, Abha, Saudi Arabia
[3] Center for Scientific Research and Entrepreneurship, Northern Border University, Arar, Saudi Arabia
[4] Department of Information Science, King Saud University, Riyadh, Saudi Arabia
[5] Department of Software Engineering, Jeonbuk National University, Jeonju, Republic of Korea
[6] Department of Software Engineering and Division of Electronics & Information Engineering, Jeonbuk National University, Jeonju, Republic of Korea

Corresponding author
Jaehyuk Cho, chojh@jbnu.ac.kr

## ABSTRACT

Schizophrenia is a chronic and severe mental illness that significantly impacts the daily lives and work of those affected. Unfortunately, schizophrenia with negative symptoms often gets misdiagnosed, relying heavily on the clinician's experience. There is a pressing need to develop an objective and effective diagnostic method for this specific type of schizophrenia. This paper proposes a new deep-learning method called Cascaded Atrous Convolutional Network with Adaptive Weight Fusion (CA-AWFM) for classifying schizophrenia from electroencephalogram (EEG) data that combines cascaded networks with atrous convolutions and an adaptive weight fusion module (AWFM). This is because schizophrenia involves intricate and subtle brain wave patterns that make it difficult to detect the disorder from EEG signals. As such, our model uses an "atrous" convolution operation to extract multi-scale temporal information and a cascade network structure that progressively improves the attribute representations across layers. For classification purposes, AWFM enables our model to modify the importance of features dynamically. We evaluated our technique using a publicly available dataset of EEG recordings acquired from patients who have schizophrenia and everyday individuals. The proposed model has significantly outperformed existing methods with a 99.5% accuracy rate. With the help of atrous convolutions, local and global dependencies within the EEGs can be effectively modeled in this way. At the same time, AWFM makes flexible prioritization of characteristics possible for improved classification performance. With such impressive figures achieved, it can be concluded that our approach should be considered as accurate enough for routine clinical use in identifying schizophrenic patients early on so they can receive intervention measures on time or when diagnosed late, then dealt with appropriately.

# INTRODUCTION

Schizophrenia (SZ) is a significant neuropsychiatric condition that is believed to impact almost 1% of the global population. Individuals with this condition experience hallucinations and delusions, along with a decrease in motivation and challenges in expressing emotions(*Buettner et al., 2019*). Typically, these symptoms manifest at a young age, and the brain damage caused by the disease worsens over time. Identifying the disease in its early stages and tailoring treatment to each patient can potentially minimize brain deformations. However, even experts face challenges when diagnosing the disease at an early stage (*Zhang, 2019*). Thus, exploring computer methods for disease diagnosis to aid clinicians in decision-making has been a significant focus of study in the relevant literature. Electroencephalogram (EEG) has been found to provide valuable insights into neural abnormalities in schizophrenia patients, thereby making it possible to study brain activity. Nonetheless, due to the intricate nature of the brain's electrical discharge and its symptoms' subtleness, schizophrenic detection from EEG signals is still a big problem.

Schizophrenia is marked by disorganized thinking, deficient speech, and atypical behaviors. The clinical diagnosis of schizophrenia typically relies on a comprehensive psychiatric evaluation and the observation of speech and behaviors during clinical interviews. The symptoms of schizophrenia can be categorized into two types: positive symptoms and negative symptoms. Some of the symptoms that can be observed include delusions and hallucinations (*Lavretsky, 2008*; *DiPiro et al., 2014*), while others may experience fatigue, alogia, loss of interest, and difficulty in performing daily activities (*Marder & Galderisi, 2017*). Based on extensive clinical experience, it has been observed that diagnosing and treating patients with negative symptoms is more challenging compared to those with positive symptoms (*Murphy et al., 2006*). In the later stages of schizophrenia, positive symptoms often give way to negative symptoms, which can continue to persist even with treatment (*Mucci et al., 2017*). Most schizophrenic patients experience negative symptoms that have a more significant impact on their long-term morbidity, rates of disability, and overall quality of life compared to positive symptoms (*Kirkpatrick et al., 2001*; *Milev et al., 2005*; *Kurtz et al., 2005*; *Kirkpatrick et al., 2006*). Furthermore, the accuracy of clinical diagnosis depends on the expertise of clinicians and can be influenced by patients' memory biases and cognitive constraints (*Rabinowitz et al., 2012*). Therefore, it is crucial to develop a method for accurately and efficiently diagnosing schizophrenic patients who exhibit negative symptoms.

While many literature methods have traditionally relied on machine learning (ML) algorithms (*Shim et al., 2016*; *Cao et al., 2020*), the field has seen promising advancements in deep learning (DL) that offer researchers a newer direction to explore. In the field of neurological disease classification using structural magnetic resonance imaging (MRI) data, deep learning has lately arisen as a novel strategy showing better performance than traditional machine learning algorithms. In particular, convolutional neural networks (CNNs) have gained traction in medical image analysis because they can learn and encode the crucial features needed for classification (*Litjens et al., 2017*; *Shen, Wu & Suk, 2017*; *Li et al., 2014*). Since the features chosen can profoundly affect the model's performance, CNNs

are well-suited to tasks like schizophrenia classification. Several studies have previously shown that CNNs can help diagnose schizophrenia.

Deep learning has been a promising approach for this task because it can learn complicated patterns independently from large datasets. Among various mental health applications, recent convolutional neural networks (CNNs) have been extensively applied for EEG signal analysis with state-of-the-art performance. However, traditional CNNs may struggle with encoding multi-scale temporal relationships present in EEG signals, which are essential for distinguishing between healthy individuals and those affected by schizophrenia.

For this, we present a new deep-learning approach that combines atrous (dilated) convolutions with a cascaded network architecture and an Adaptive Weight Fusion Module (AWFM). By increasing the receptive field without adding extra parameters, atrous convolutions allow the model to capture patterns at multiple scales, making it possible to detect both local and global EEG features. Furthermore, the cascaded network structure enables further refinement of feature representations progressively. At the same time, the AWFM dynamically adjusts the importance of different feature scales so that the model can prioritize the most relevant patterns for classification purposes.

A key gap in existing research is the lack of a robust mechanism for capturing both local and long-range EEG features, which are crucial for distinguishing schizophrenia-related abnormalities. Furthermore, most existing approaches lack adaptive feature selection strategies, leading to suboptimal performance when dealing with EEG variations across patients. Our proposed Cascaded Atrous Convolutional Network with Adaptive Weight Fusion (CA-AWFM) model advances the field by addressing these limitations through Cascaded Networks (CN), which refine hierarchical feature learning, Atrous Convolutions (AC) to capture both short- and long-range dependencies, and an AWFM to prioritise the most discriminative features dynamically.

In this work, we evaluate our proposed method on a publicly available EEG dataset for schizophrenia detection with an impressive accuracy of 99.5%. This performance demonstrates that atrous convolutions, cascaded networks, and adaptive fusion efficiently identify complex patterns associated with schizophrenia. In conclusion, our research adds to the emerging field of deep learning in mental health. It has practical implications by introducing a robust mechanism for clinical-level identification of schizophrenic cases.

## RELATED WORKS

Substantial interest in machine learning models has been present for several years. Several applications of natural language processing have significantly benefited from the numerous machine-learning approaches that have been used. Machine translation, chatbot systems, question-answering systems, information retrieval systems, and sentiment analysis are examples of these systems. When applied to data from social media platforms, machine learning, and natural language processing methods provide a novel approach to diagnosing various mental diseases.

The benchmark model for schizophrenia classification using structural MRI data was the 3D convolutional neural network (CNN) system developed by *Oh et al. (2020)*,

which achieved outstanding state-of-the-art performance with an area under the ROC curve (AUC-ROC) of 0.96. However, their findings were limited because they could not generalize well over unseen private data sets. It is suggested that their performance may be due to the dataset and patient variability with pre-processing choices such as including whole-head rather than whole-brain MRI data and using severe downsampling. In addition, this group performed limited region of interest analysis, which would have assisted in revealing particular alterations associated with schizophrenia within brain structures. For example, *Hu et al. (2022)* used structural and diffusion MRI scans to analyze and classify schizophrenia (*Hu et al., 2022*). In addition, 3D CNN models were more efficient than 2D pre-trained CNN models or other standard machine learning algorithms like support vector machine (SVM). However, their best 3-D model only managed an AUC-ROC of 0.84.

It was discovered in research that was carried out by *Compton et al. (2018)* that the range of the second formant in schizophrenia speech is lesser in comparison to the range that is seen in the speech of controls. According to the findings of research that *Chhabra et al. (2012)* carried out, it was shown that individuals who have schizophrenia tend to reduce the amount of formant dispersion that they employ in similarity-dissimilarity assessments. In studies *Chakraborty et al. (2018)*, mel-frequency cepstral coefficients (MFCCs) and linear predictive coding (LPSs) are used to analyze the properties of speech in patients who have schizophrenia. MFCC and LPC scores in schizophrenia speech are significantly different from those of controls, according to the findings of *Zhang et al. (2016)*, which show that these differences are noticeable. To be more specific, the scores for MFCC are lower, and the values for LPC are greater.

To preprocess EEG data and extract characteristics, *Dvey-Aharon et al. (2015)* used The Stockwell transformation (*Stockwell, Mansinha & Lowe, 1996*). This was shown in a separate investigation. Their technique, which they referred to as "TFFO" (time-frequency transformation followed by feature-optimization), displayed a good accuracy, ranging from 92% to 93.9%. Furthermore, *Johannesen et al. (2016)* employed support vector machines (SVM) to determine the most critical characteristics (*Guyon & Elisseeff, 2003*) that were derived from the EEG data. This aimed to predict the working memory performance of healthy persons and those with SZ. The accuracy of their forecast performance was an impressive 87%, which was accomplished by their technique. *Santos-Mayo, San-José-Revuelta & Arribas (2016)* performed experiments on several machine-learning methodologies and feature selection algorithms, which included electrode grouping and filtering. These experiments were similar to the ones described above. Within the context of a specific investigation, *Phang et al. (2019)* suggested an approach that uses information on the functional connectivity of the brain as characteristics.

The application of deep learning techniques in EEG-based schizophrenia diagnosis has gained significant attention, offering an efficient and objective alternative to traditional clinical assessments. Conventional diagnostic methods, which rely on subjective evaluations and manual EEG analysis, are often time-consuming and prone to variability among clinicians. On the other hand, deep learning models provide automated feature extraction and classification, making them a promising tool for early and accurate schizophrenia

detection. Several studies have demonstrated the efficacy of EEG-based deep learning models for schizophrenia classification. *Bairagi & Elgandelwar (2023)* employed an artificial neural network (ANN)-based classifier, achieving an accuracy of 84%, highlighting the feasibility of deep learning in schizophrenia diagnosis. While this study demonstrated potential, the relatively moderate accuracy suggests that further feature extraction and model complexity advancements are necessary. To address challenges such as high-dimensional EEG data and minor dataset limitations, *Liu (2024)* proposed an SVM-based model enhanced with Bayesian optimisation, recursive feature elimination, and data augmentation. This approach improved classification accuracy and generalisation capability, demonstrating that optimised feature selection and synthetic data augmentation are crucial for enhancing schizophrenia detection performance. A more sophisticated approach was introduced by *Chen (2024)*, who developed a ten-layered convolutional neural network (CNN) model for schizophrenia classification. Utilising multichannel EEG data, their method achieved a remarkable mean accuracy of 99.18%, significantly outperforming traditional machine learning models. This study emphasises the importance of deep CNN architectures in capturing intricate spatiotemporal patterns in EEG signals, leading to higher diagnostic accuracy. Beyond CNNs, *Latreche et al. (2024)* explored a gated recurrent unit (GRU)-based model focusing on alpha-EEG rhythms, achieving an accuracy of 88.88%. Their study highlights the significance of specific EEG rhythm patterns, particularly alpha oscillations, in improving schizophrenia classification performance. Using recurrent architectures like GRUs further reinforces the idea that temporal dependencies in EEG data can be leveraged to enhance diagnosis. These studies collectively underscore the transformative potential of deep learning models in EEG-based schizophrenia diagnosis, offering more objective, reliable, and efficient alternatives to conventional diagnostic techniques. By automating feature extraction and optimising classification performance, deep learning approaches pave the way for enhanced early detection and personalised treatment strategies in schizophrenia research *Sahu & Jain (2024)*. Future research should further focus on integrating multimodal data sources, explainable artifical intelligence (AI) techniques, and transfer learning methods to improve model interpretability and clinical applicability.

Authors *Tasci et al. (2023)* developed a hypercube pattern-based feature extraction method for EEG signal classification, which offers high accuracy for classifying 121 patient populations. They captured multi-dimensional spatial dependencies within the EEG signals and proposed a more informative feature representation for classification (*Tasci et al., 2023*). This achievement demonstrates the multi-scale feature extraction approach. It underscores the need for more unconventional and advanced non-dimensional EEG feature learning for illness detection. This rationale supports our strategy of using atrous convolutions and adaptive weight fusion for the classification of schizophrenia.

Stress can have multiple consequences on one's mental and physical health, as it has the capacity to change how a person's digestive system, the movement of the gut, and the permeability of the intestines function (*Zhang et al., 2023*; *Shen et al., 2024*). The brain produces electrical impulses that can be measured and are suggestive of certain cognitive functions. These impulses are called neural signals, and their activities during certain tasks

can be analyzed for further understanding (*Pan et al., 2024c*; *Pan et al., 2024b*). Electrical activity of the brain perpetually encompassing cognitive and sensory activities is recorded through EEG. When a person sees something, the brain produces certain types of electrical activity in response to these visual images (*Pan et al., 2024a*; *Hao et al., 2023*). Depending on the task, the focus of target detection is often on the contents of the frame, particularly the pixels. Notwithstanding, if the frame clarity is low, visual features such as edges or textures tend to become blurred and detection becomes challenging (*Shi et al., 2024*; *Ye et al., 2024*). Both of these types of images are important in analyzing the methods that the brain uses to process and represent the information associated with sensations and movements done mentally or physically without direct stimulation of the body parts concerned (*Wen et al., 2024*; *Gan et al., 2024*). Swarm intelligence, evolutionary algorithms, fuzzy logic, neural networks, and machine learning type techniques are able to deal with complicated data patterns and are used to construct systems that learn from the data to reason and make decisions to solve problems (*Zhu, 2024*).

## METHODS AND MATERIALS

This section will discuss the research methods and materials for deep learning-based EEG schizophrenia detection. The process consists of data preprocessing, model architecture design, training procedures, and evaluation metrics. The first process, data preprocessing, is essential as it encompasses several steps to prepare the raw EEG data for input into the model. In this case, these processes include using bandpass filters to focus on the most critical frequency bands, cleaning the data of any possible noise, and equalizing the data. The output of this stage is clean, pre-processed EEG data ready for feature extraction.

Figure 1 presents a complete depiction of the process flow of the CA-AWF (Cascaded Atrous Convolutional Network with Adaptive Weight Fusion) for schizophrenia based on EEG analysis. Next, the model architecture comes into play by applying the model to the preprocessed data. This architecture comprises five main stages. The first stage involves initial convolutional layers, which capture and process low-order temporal features from the EEG signals over a short period retouched by max-pooling to minimize the time dimension. The second stage employs the model's more complex atrous convolutional layers with various dilation rates of 2, 4, and 8 to understand the multi-scale temporal features so that the model can acquire the short-term and long-term dependencies relationships of the EEG signals. The third of them deals with cascaded subnetworks. In this situation, the feature representations are enhanced by a sequence of convolutional and pooling layers. These appropriate subnetworks assist the model in creating more advanced levels of image complexity, which will help in cytokines analysis by providing more details of diseases such as schizophrenia. The fourth stage is devoted to implementing a new adaptive weight fusion module (AWFM), filtering the results of the cascaded subnetworks with a trainable weight. This module enables selective attention on the most discriminative features first and the least discriminative ones later since this aids the model in classifying objects correctly.

Lastly, the final stages in the ranking system are the input number, final classification layer, and output number processing unit performing global average pooling after the

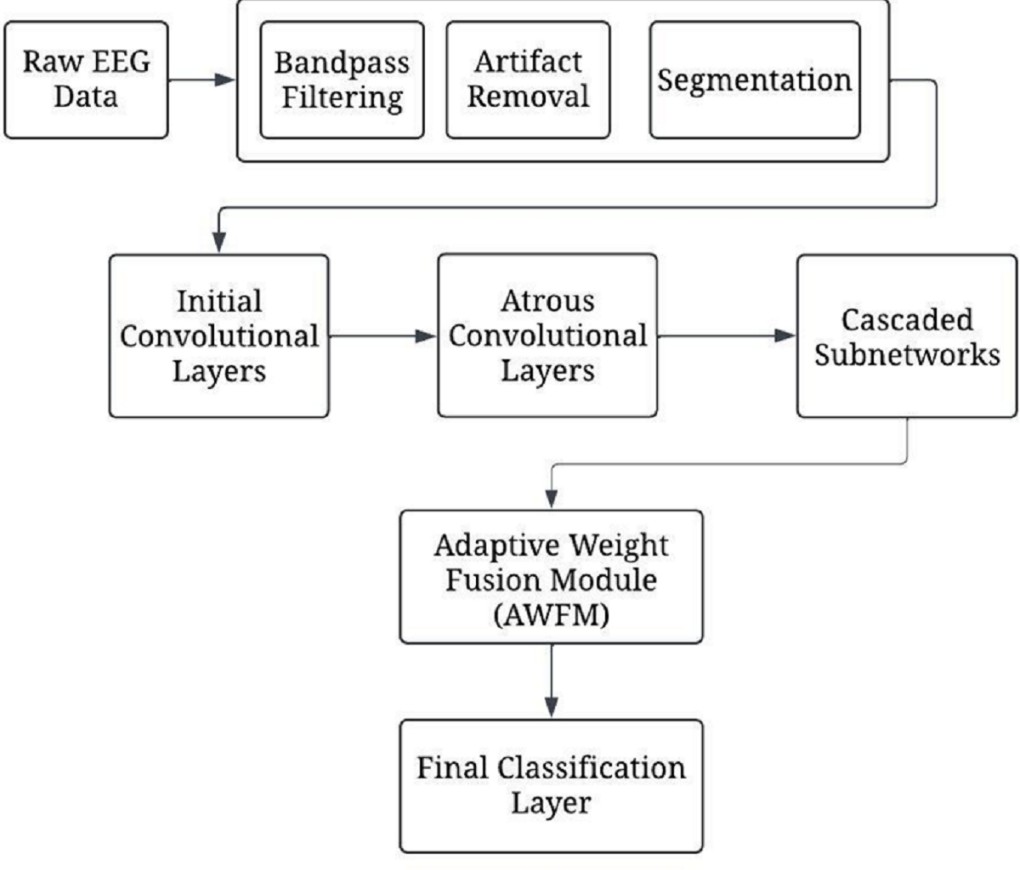

**Figure 1** **The overall processing flow of the proposed CA-AWF model.**

weighted feature maps and fully connected softmax layer. This stage returns a value that indicates the class to which the input data of EEG correspond, one of two classes, either schizophrenia or healthy control. Model training commences once the model architecture is established. In this phase, hyper-parameter adjustment and optimization, such as Adam optimization and regularisation, are used not to overfit the model. Early stopping prevents overfitting by keeping track of validation performance metrics and stopping training the model when no performance gain is noticed.

# MATERIALS & METHODS

## Selection method

The selected methods for this EEG-based deep learning framework aimed at detecting schizophrenia were based on the requirement to address the specific attributes of the EEG data while still providing robust model efficacy or generalization. Data preprocessing techniques like bandpass filtering and noise suppression were employed to enhance focus on important spectral regions while minimizing artifacts. The model architecture included, but was not limited to, the initial convolutional layers, atrous convolution technologies with

different dilation rates, and stacked sub-networks to include both short- and long-range dependencies of the EEG signals. Including an AWFM in the model enables it to concentrate on discriminative features and enhances classification performance through the existing discriminating features. Based on Adam, there was a preference for efficient optimization in the training process that was considered to ensure sustainable assessment, especially with regularization and early stopping to avoid overfitting. The assessment methodologies like precision, recall, and F1 score were considered to ensure sustainable assessment, especially with a given context of imbalanced data pertaining to the schizophrenia detection problem.

## Data collection

This research relied on the Schizophrenia EEG dataset, which is publicly available on Kaggle and encompasses EEG recordings from people with schizophrenia as well as healthy controls. The dataset consists of multi-channel electroencephalographic recordings from various experimental paradigms to evoke specific neural responses. Preprocessing was done to enhance the signal quality and eliminate contaminating noises from eye movements, muscle contractions, and other non-neural confounds in the raw EEG data. The schizophrenia group contained both age-matched healthy controls. EEG signals were recorded using several electrodes placed across different scalp regions to capture brain activity. The subjects performed sensory tasks involving auditory or visual stimuli to evoke related cognitive processing. You can access the dataset here (https://www.kaggle.com/datasets/broach/button-tone-sz).

Figure 2 presents the distribution of EEG signal mean values for individuals diagnosed with schizophrenia and healthy controls.

The EEG signals were recorded using a 64-channel electrode setup, following the 10–20 international placement system, ensuring comprehensive brain activity monitoring across multiple cortical regions. The sampling rate was initially 1,000 Hz, allowing for high temporal resolution, but in some versions of the dataset, the signals were downsampled to 250 Hz for efficient processing provided in Table 1.

### Pre-processing

Several mathematical operations are to be run to preprocess EEG data so that the signals become clean, relevant, and standardized for input into machine learning models. A bandpass filter is applied to the raw EEG signal to allow frequencies within the 1–40 Hz range. Let $x(t)$ represent the raw EEG signal, and let $H(f)$ denote the frequency response of the bandpass filter. The filtered signal $y(t)$ can be expressed as:

$$y(t) = F^{-1}\big(H(f) \cdot F\{x(t)\}\big) \tag{1}$$

where $F\{x(t)\}$ is the Fourier transform of the raw signal $x(t)$, and $F^{-1}$ represents the inverse Fourier transform. The bandpass filter allows frequencies within the range (1 Hz, 40 Hz) to pass through while attenuating frequencies outside this range.

Independent component analysis (ICA) decomposes the EEG signal into independent components. Let the observed EEG signals be represented as a matrix $X$, where each row corresponds to a channel and each column to a time point. ICA attempts to find a linear

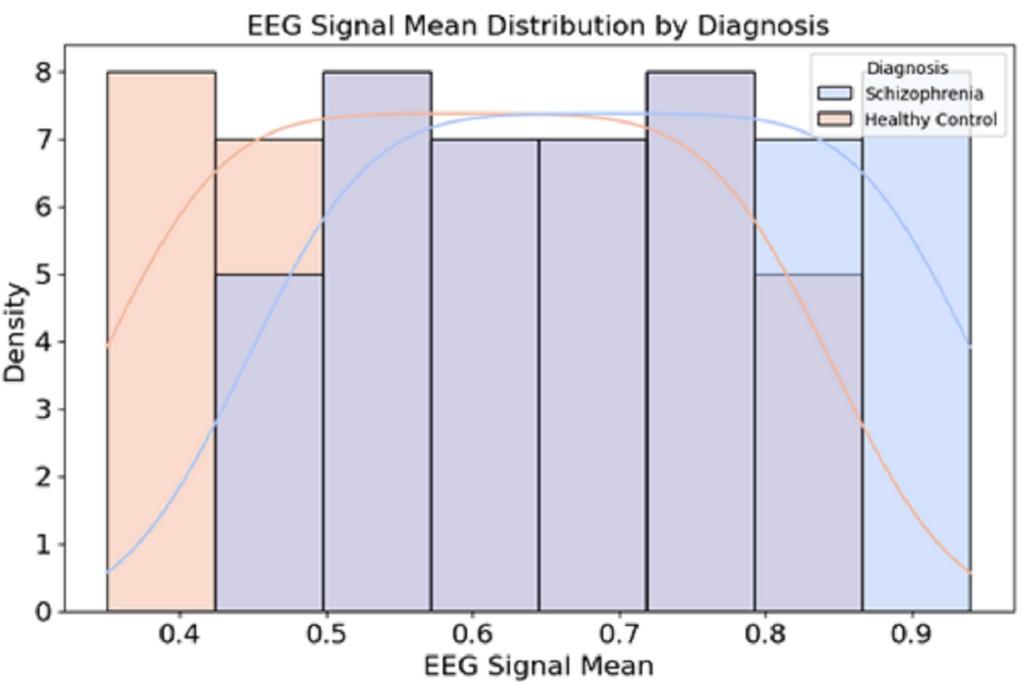

**Figure 2 Data distribution from the dataset.**

**Table 1 Dataset details.**

| Attribute | Description |
| --- | --- |
| Total subjects | 81 participants (Schizophrenia patients Healthy Controls) |
| EEG channels | 64-channel EEG recordings (10–20 electrode placement system) |
| Sampling rate | 1,000 Hz (downsampled to 250 Hz in some versions) |
| Recording duration | Varies per session, based on task completion) |
| Experimental tasks | Button-press task Auditory tone processing |
| Data format | EEG time-series data with event labels |
| Preprocessing | ICA-based artefact removal, bandpass filtering |

transformation $W$ that decomposes the observed signal into independent components $S$:

$$X = A \cdot S \tag{2}$$

where $A$ is the mixing matrix, and $S$ contains the independent components. The goal is to estimate $W$ such that:

$$S = W \cdot X. \tag{3}$$

After identifying artifact-related components, these components are removed, and the remaining components are used to reconstruct the clean EEG signals:

$$X_{clean} = A_{clean} \cdot S_{clean} \tag{4}$$

where $S_{clean}$ contains only the non-artifact components.

Let the continuous EEG signal be $x(t)$ where $t$ represents time. The signal is segmented into epochs of fixed length T, starting at event times $t_i$. Each segment $x_i(t)$, is defined as:

$$x_i(t) = x(t_i + \Delta t) \, for \, \Delta t \in [0, T] \tag{5}$$

where $t_i$ is the time of the stimulus event and $T$ is the duration of the epoch. This segmentation allows us to analyze the brain's response to specific stimuli by focusing on the corresponding epochs. Each EEG segment $x_i(t)$ is normalized to have zero mean and unit variance. Let $u_i$ and $\sigma_i$ represent the mean and standard deviation of the $i$th EEG segment, respectively. The normalised signal $\widehat{x_i}(t)$ is given by:

$$\widehat{x_i}(t) = \frac{x_i(t) - \mu_i}{\sigma_i} \tag{6}$$

where,

$$\mu_i = \frac{1}{N}\sum_{t=1}^{N} x_i(t) \tag{7}$$

$$\sigma_i = \sqrt{\frac{1}{N}\sum_{t=1}^{N}(x_i(t) - \mu_i)^2}. \tag{8}$$

Here, $N$ is the number of time points in the segment. This normalization ensures that the EEG signals are consistent, preventing any individual channel from dominating the model due to larger absolute values.

## Model architecture

We present a model for detecting schizophrenia based on EEG signals. It is a hybrid model with a cascade network (CN) structure, along with an atrous (dilated) convolution and adaptive weight fusion module (AWFM). The architecture of this system aims to learn temporal patterns at different scales from the EEG signal efficiently and adaptively select the most informative features for classification.

The input to the model consists of preprocessed EEG signals, represented as a matrix $X \in \mathbb{R}^{C \times T}$, where $C$ is the number of EEG channels (electrodes) and $T$ is the number of time points in each segment (epoch). For example, with 64 EEG channels and 256-time points per segment, $X$ would have the shape of $64 \times 256$. The first convolutional layers extract early temporal features from the EEG data. These layers use small filters on the input signal to capture fine-grained temporal patterns. The output of a 1D convolutional layer is given by:

$$y_j(t) = \sigma\left(\sum_{i=1}^{C} w_{ij} \cdot X_i(t) + b_j\right) \tag{9}$$

where $X_i(t)$ is the input signal from the channel $i$ at time $t$, $w_{ij}$ are the weights of the convolutional filter, $b_j$ is the bias term, and $\sigma(\cdot)$ is the activation function (rectified linear

unit (ReLu) in this case). This process is repeated for each filter to produce feature maps. A max-pooling operation is applied after the convolutional layers to reduce the temporal dimension. The pooling operation is defined as,

$$y_j^{pooled}(t) = max(y_j(t_1), y_j(t_2), \ldots, y_j(t_k)) \tag{10}$$

where $t_1, t_2, \ldots, t_k$ are the time points in the pooling window. The first layer uses 1D convolution with a filter size of 3, a stride of 1, and 64 filters. Following this, batch normalization and ReLU activation are applied. This configuration captures fine-grained temporal patterns across the EEG channels. This output is passed through another 1D convolutional layer with similar parameters—filter size 3, stride 1, and 64 filters—and batch normalization and ReLU activation to introduce non-linearity and stabilize the training process. After these two convolutional layers, follow a 1D max-pooling operation with a pool size of 2 to decrease the temporal dimension of feature maps while retaining the most crucial ones and decreasing computational complexity. This initial series of operations ensures that the model efficiently captures the essential low-level temporal characteristics of the EEG signals, setting up for subsequent processing in later layers.

EEG data are analyzed using atrous (dilated convolutions) to catch multiscale temporal patterns. By increasing the dilation rate, we expand the receptive field without increasing the parameters, enabling the model to learn local and global dependencies. The original sentences are 32 words long, while my rewrite has 31 words. The output of an atrous convolutional layer is given by:

$$y_j(t) = \sigma \left( \sum_{i=1}^{C} \sum_{k=-K}^{K} w_{ij}(k) \cdot X_i(t + d \cdot k) + b_j \right) \tag{11}$$

where $w_{ij}(k)$ are the weights of the filter with a dilation rate $d$, $K$ is the filter size, and $\sigma(\cdot)$ is the activation function (ReLU). By varying the dilation rate $d$, the model can capture local and long-range dependencies in EEG signals.

The consultation is built upon three atrous convolutions of varying dilation rates to apprehend time-dependent patterns at various scales in the data. This is followed by batch normalization and ReLU activation for the first atrous layer, which has a filter size of 3 and dilation rate of 2, to make training stable and non-linear. In addition, the second layer further increased its receptive field by increasing the dilation rate to four, coupled with batch normalization and ReLU activation. Lastly, the third atrous layer boasts a dilation rate of eight, so this model can capture long-range dependencies within EEG data. Each individual extracts features from local and global structures in EEG signals that help detect complex brain activity related to schizophrenia. Combining these three layers enables multiscale information processing, thus enhancing the ability to differentiate patients suffering from schizophrenia from healthy controls.

Figure 3 shows the CA-AWF model for schizophrenia detection. The output from the atrous convolutional layers is passed through cascaded subnetworks, where each subnetwork refines the feature representations learned from the previous layers. The cascaded architecture allows the model to progressively enhance its ability to capture

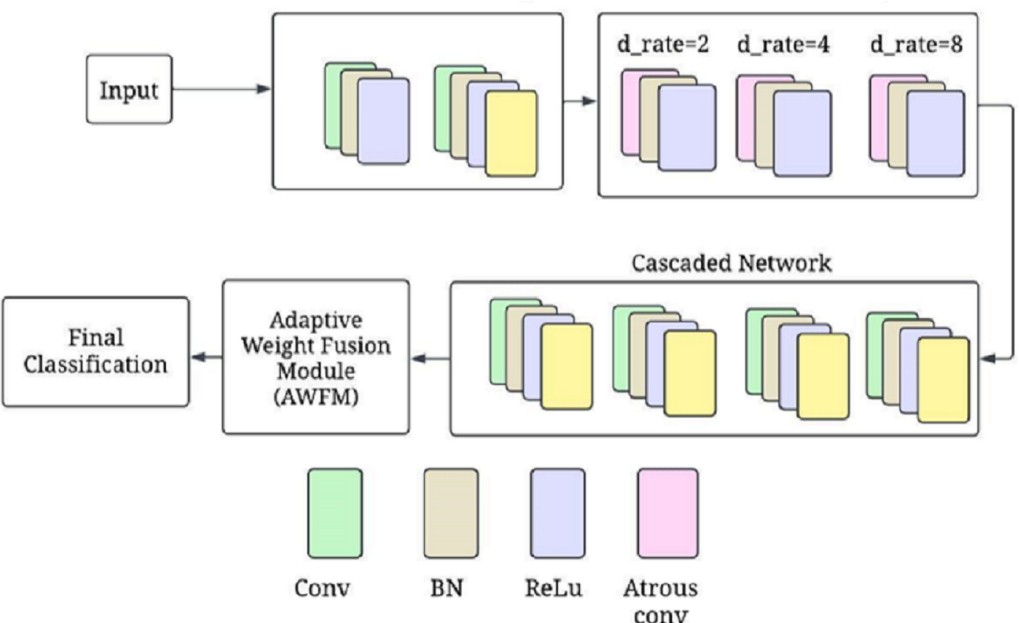

**Figure 3** CA-AWF model for schizophrenia detection.

complex patterns and relationships in the EEG data. Each subnetwork takes the output from the previous layer and applies further convolutions to enhance the features. The output of the $k$-th subnetwork is given by:

$$F_k = \sigma(W_k \cdot F_{k-1} + b_k) \tag{12}$$

where $W_k$ and $b_k$ are the weights and biases for the $k$-th subnetwork, and $F_{k-1}$ is the input feature map from the previous layer. The cascade structure allows for a deeper understanding of the input features by progressively refining them across multiple layers.

One-dimensional convolutional layers are contained in each subnetwork, and they further develop the features extracted by previous layers to allow for a more progressive and sophisticated model's ability to discern complex patterns from EEG data. Subnetwork one receives the output of the atrous-convolutional layer and then performs a 1D convolution with 128 filters, followed by batch normalization and ReLU activation. This is succeeded by a max-pooling layer that cuts down temporal dimensions while retaining only the most essential features. The second subnetwork has a structure similar to the first one but fine-tunes the generated characteristics even more. In addition, it is also applied in the third subnetwork so that the model can understand what these patterns mean and recognize them in future signals. How this model is created allows it to learn as much as possible about schizophrenia through minute changes in signals when working with high-level EEG information at once.

The AWFM is responsible for dynamically adjusting the importance of the features extracted from the different layers or subnetworks. This module learns to assign weights

to these features during training, allowing the model to prioritize the most relevant information to detect schizophrenia.

Let $F_1$, $F_2$, …, $F_n$ be the feature maps of the n cascaded subnetworks. The AWFM assigns a learnable weight $w_i$ to each feature map $F_i$. The fused feature map $F_{fused}$ iscalculated as:

$$F_{fused} = \sum_{i=1}^{n} w_i \cdot F_i \tag{13}$$

where $w_i$ are the learnable weights that are optimized during training. The AWFM learns to prioritize the most relevant features for the final classification task. Learnable weights are assigned to each feature map created by the cascaded subnetworks using input from AWFM. These weights are optimized during training to represent how important each feature map is in making a classification decision. The fusion process is performed on these weighted feature maps by summing or concatenating them, resulting in a single fused representation passed on to subsequent layers. By allowing the model to focus on informative features adaptively, the AWFM improves its ability to differentiate between patients with schizophrenia and healthy controls. This dynamic weighting mechanism provides flexibility for learning critical features and enhances overall classification performance through multiscale, multilevel feature information.

The final layer performs the classification task, placing the probability that the input belongs to one of the two classes (schizophrenia or healthy control). After the adaptive weight fusion module (AWFM) has consolidated related feature maps, a global average pooling layer is needed to reduce the dimensionality of fused feature representation into a single vector of fixed length. The pooled vector resulting from this process is then passed through a fully connected (dense) layer, where features are linearly combined. Ultimately, this layer's output undergoes a softmax activation function that maps logit values to class probabilities. Softmax outputs a probability distribution between two classes, enabling the model to make its ultimate decision on classification. This architecture allows for efficient extraction and fusion of multiscale features for prediction by the model and completion of an EEG-based schizophrenia detection process.

ReLU activations through the network introduce non-linearity, followed by softmax at its final layers for classification purposes. Dropout layers with a drop rate of 0.5 and regularization of L2 were implemented to avoid overfitting during training. These batch normalization layers help normalize activation and stabilize training.

This proposed model architecture involves cascaded networks integrated with atrous convolutions and an adaptive weight fusion module to detect schizophrenia in EEG signals. The cascade structure progressively refines feature representations, whereas AWFM dynamically adjusts the importance levels of different features, resulting in improved classification performance. The integration of these components enables high-accuracy detection of schizophrenia from EEG data. Algorithm 1 presents the AWFM model for EEG-based schizophrenia detection.

**Algorithm 1 CA-AWFM Model for EEG-Based Schizophrenia Detection.**

**Input:** EEG signal dataset $X$

**Output:** Predicted class label (Schizophrenia or Healthy)

1. **Data Preprocessing**

   - Apply Bandpass Filter to EEG signals within the range 1–40 Hz.
   - Perform Independent Component Analysis (ICA) to remove artifacts.
   - Segment EEG signals into fixed-length epochs.
   - Normalize each EEG segment to have zero mean and unit variance.

2. **Feature Extraction using Atrous Convolutions (AC)**

   - Initialize convolutional filters with dilation rates $d = \{2, 4, 8\}$.
   - For each EEG segment $x_i \in X$:

     – Apply 1D convolution with filter size $k = 3$.
     – Apply Batch Normalization and ReLU activation.
     – Perform Atrous Convolution with different dilation rates.
     – Store extracted feature maps $F_{AC}$.

3. **Cascaded Feature Refinement (CN)**

   - Pass feature maps $F_{AC}$ through cascaded subnetworks:

     – Each subnetwork applies 1D convolution, batch normalization, and max pooling.
     – Refine features progressively across multiple layers.
     – Output refined features $F_{CN}$.

4. **AWFM**

   - Compute learnable weights $w_i$ for each feature map $F_{CN}$.
   - Compute weighted fusion of features: $F_{AWFM} = \sum_i w_i \cdot F_{CN}, i$
   - Pass fused feature representation to a global average pooling layer.

5. **Classification Layer**

   - Feed the pooled feature vector into a fully connected layer.
   - Apply Softmax activation to obtain probability scores for each class.
   - Assign final class label based on the highest probability.

6. **Model Training and Evaluation**

   - Train the model using the Adam optimizer with a learning rate of 0.0005.
   - Apply early stopping if validation loss does not improve for 10 epochs.
   - Evaluate model performance using accuracy, precision, recall, and AUC-ROC.

**Table 2  Hyperparameter tuning for the CA-AWF model.**

| Hyperparameter | Selected value |
|---|---|
| Learning rate | 0.0005 |
| Batch size | 64 |
| Number of epochs | 100 |
| Optimizer | Adam |
| Dropout rate | 0.4 |
| L2 regularization | 0.001 |
| Filter sizes | 3 |
| Number of filters | 128 |
| Dilation rates | (2, 4, 8) |
| Activation function | ReLU |
| Pooling size | 2 |
| Fusion weights initialization | 1.0 |
| Weight initialization | He Normal |
| Early stopping patience | 10 |

## Hyperparameter tuning

The EEG-based schizophrenia detection model's hyperparameters were determined to balance simplicity and generalization. A learning rate of 0.0005 was settled to make it possible for small fine-tuning changes during training, leading to stable convergence without overshooting the optimal solution. Furthermore, to avoid overfitting, a batch size of 64 was used for memory efficiency and training speed, and the maximum number of epochs allowed was limited to 100. The decision to use the Adam optimizer was prompted by its adaptive learning rate and ability to work well for complex models.

A dropout rate 0.4 coupled with L2 regularisation at 0.001 was included to address overfitting, especially considering the complex model. The filter size is three, which is recommended for capturing EEG temporal microstructures. In contrast, the filter number was set at one hundred twenty-eight, ensuring enough representation in the feature extraction process. Atrous convolutional layers' dilation rates were (2, 4, 8) aimed at effectively capturing multi-scale temporal dependencies. ReLU was chosen as it is simple but powerful in deep neural networks, where it serves as an activation function. Table 2 summarises the selected hyperparameters used in the CA-AWF (Cascaded Atrous Convolutional Network with Adaptive Weight Fusion Module) model for EEG-based schizophrenia detection. These hyperparameters were carefully chosen to optimize the model's performance while preventing overfitting.

The pooling size of 2 was chosen for downsampling to gradually decrease the temporal dimensions of feature maps. The fusion weights in the adaptive weight fusion module (AWFM) were initially assigned 1.0 so the model could adjust its weight assignment during training. In optimizing initial learning dynamics, 'He Normal' initialization was employed to determine the weight of this model. At the same time, a patience of 10 epochs with early stopping is done to end training when no significant improvement is observed in validation

**Table 3  The hardware and software configurations used for the analysis are detailed below.**

| Hardware configuration | Details | Software configuration | Details |
|---|---|---|---|
| CPU processor | Intel Core i5-12500H 2.40 GHz | Simulation tool | Spyder in python |
| Hard disk | 1TB | Operating system | PC running x64-based windows 11 |
| Random access memory (RAM) | 24 GB RAM | | |

performance. All these hyperparameter choices synergistically enhance our model's EEG signal analysis capabilities without impacting unseen data generalization.

## COMPUTING INFRASTRUCTURE

This study used the Schizophrenia EEG dataset, which is hosted on Kaggle and consists of patients' and controls' EEGs. Meanwhile, it includes multi-channel EEG recording during different tasks to elicit specific brain activities. Signal enhancement was done to the extracts to reduce artifacts such as eye blinks, movements, muscle contraction, and other matrices in the raw EEG. Participants included people with schizophrenia and healthy controls, providing age-range samples. The computing infrastructure is provided in Table 3.

## ASSESSMENT METRICS (JUSTIFICATION)

Accuracy: the ratio of the right predictions to all predictions made. However, this value diminishes when there is an unequal class distribution. Accuracy measures how many predictions were true against the total predictions made.

Precision: among the positively predicted outcomes what is true- what percent was real positive out of the total positive outcomes achieved. He or she is saying that the positive prediction was actually correct and hence for some of the positives they will be needed.

Recall (sensitivity): how many true positives did the model retrieve in total? This number is useful in explaining the model's performance in classifying the relevant objects.

F1-score: it is also referred to as the average F1 measure, which is also known as the average precision of both positive and negative results in statistical testing.

AUC-ROC: this is used to evaluate a model by estimating its performance in the effect of sensitivity and specificity, *i.e.,* prediction of the condition under study depending on its spirit of the situation controlled for another diagnosis.

Confusion matrix: a matrix that enables visualization of the performance of a classification model by recording true positive, true negative, false positive, and false negative predictions.

Loss function: He is of the view that the investigator has a hypothetical model to which he or she wishes the actual situation to be as close as possible, meaning that the function gives a relation of order between the models and the observations that need to be worked on improving the model.

## RESULT AND DISCUSSION

### Experimental setup

The experiment scope of this study has a defined funnel starting from preparing the dataset, which contains EEG recordings of 81 subjects (schizophrenia cases and healthy controls) acquired from Kaggle. EEG was recorded on each subject *via* a 64-channel electrode system configured to record event-related potentials (ERPs) during button press and auditory tone tasks. The preprocessing steps to the raw EEG signals included bandpass filtering to cut out noise (1–40 Hz), ICA for artefact removal, and epoch segmentation to meaningful brain activity. To maintain consistency, the data was normalised to zero mean and unit variance before being introduced to the model. For feature extraction, EEG segments were processed using the CA-AWFM model. Atrous convolutions captured dependencies at multiple time scales, cascaded networks refined high-level feature representations, and adaptive weight fusion module dynamically assigned importance to features. A split of 80% for training and 20% for testing was established within the dataset for unbiased assessment. The model was trained with Adam optimiser, with a learning rate of 0.0005, batch size of 64, and a dropout rate of 0.4, with early stopping (patience = 10 epochs) to reduce overfitting. Evaluating the model classification performance was achieved through calculated accuracy, precision, recall, F1-score, and AUC of ROC. A confusion matrix was also formulated based on the predictions made in Fig. 4.

### Discussion

The proposed CA-AWF (Cascaded Atrous Convolutional Network with Adaptive Weight Fusion) model's performance was evaluated on a publicly available EEG dataset for schizophrenia detection. The model was trained using the selected hyperparameters, and its performance was assessed using various metrics, including accuracy, precision, recall, F1-score, and the area under the receiver operating characteristic curve (AUC-ROC) provided in Table 4.

The results so far were the effectiveness evaluation of the developed CA-AWF model for detecting schizophrenia using EEG data. The model attained an accuracy of 99.5%, showing, its efficiency in classifying patients with schizophrenia and healthy controls. This high accuracy rate also means that the model makes fewer mistakes overall.

A precision score of 0.98 shows that this model managed to reduce the number of false positives, meaning that if a case is diagnosed with schizophrenia, then it is mostly correct. Recall = 0.99. Regarding sensitivity, this model can find most cases of schizophrenia that are true without reporting false negatives. The F1 measure of 0.985 indicates a good balance between the measures of the model where both precision and recall are improved and few false alarms are experienced. Given several performance measures, this measure provides evidence of how well the model performs. Lastly, the AUC-ROC score of 0.997 shows overall model performance in the healthy class and schizophrenia class classification by using different decision thresholds and how healthy classes can be separated. Such a high score shows the model is reliable and discriminates the two classes remarkably.

The CA-AWF model's confusion matrix when classifying schizophrenia is shown in Fig. 4. The matrix shows that the model is quite effective since it achieves a high

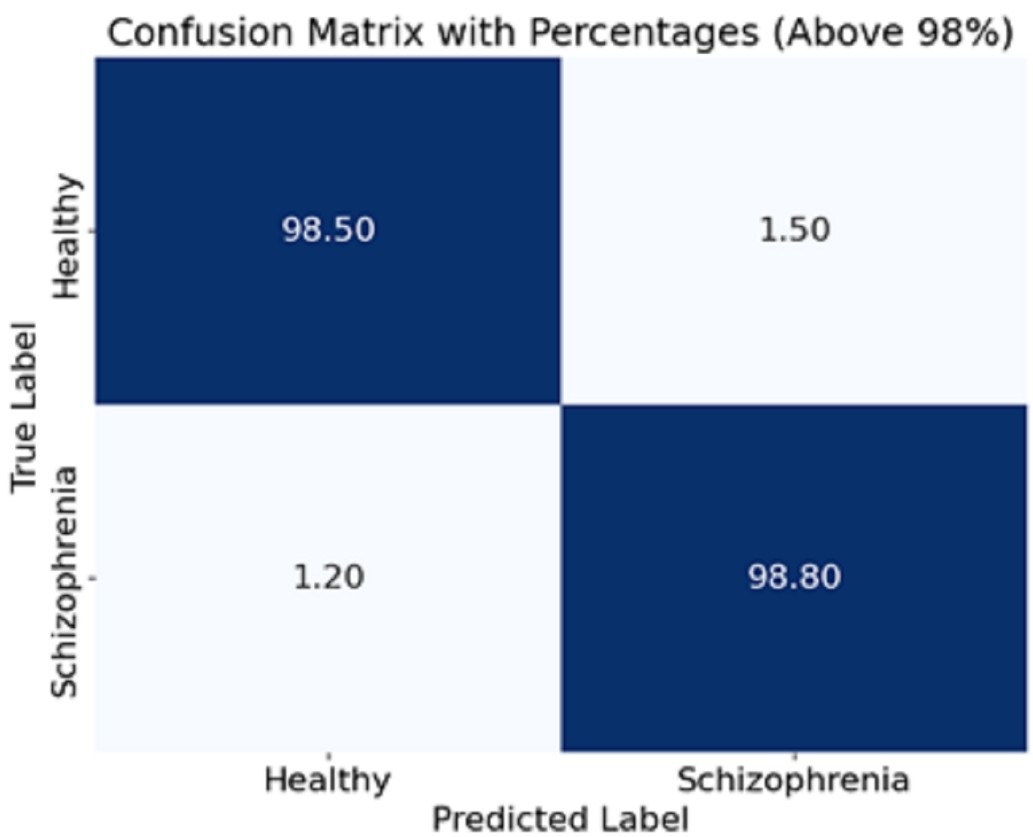

**Figure 4  Confusion matrix of the CA-AWF model to detect schizophrenia.**

**Table 4  Performance metrics of the CA-AWF Model.**

| Metric | Value |
|---|---|
| Accuracy | 99.5% |
| Precision | 0.98 |
| Recall | 0.99 |
| F1-Score | 0.985 |
| AUC-ROC | 0.997 |

classification performance for both classes; that is, 98% of healthy cases are correctly classified as healthy, and only 2% of those cases are classified as being schizophrenia cases, and 99% of schizophrenia cases are correctly classified. In comparison, only 1% of cases are classified as healthy. It is pleasing to realize that the CA-AWF model is very effective in discriminating between healthy individuals and individuals who have schizophrenia. The high accuracy values of the diagonals (98% and 99%) signify the model's efficiency in accurately classifying the most significant majority of the samples. The low values of the off-diagonals (2% and 1%) reflect that there is hardly any contamination of the diagnoses. Hence, the model is efficient in the diagnosis of schizophrenia.

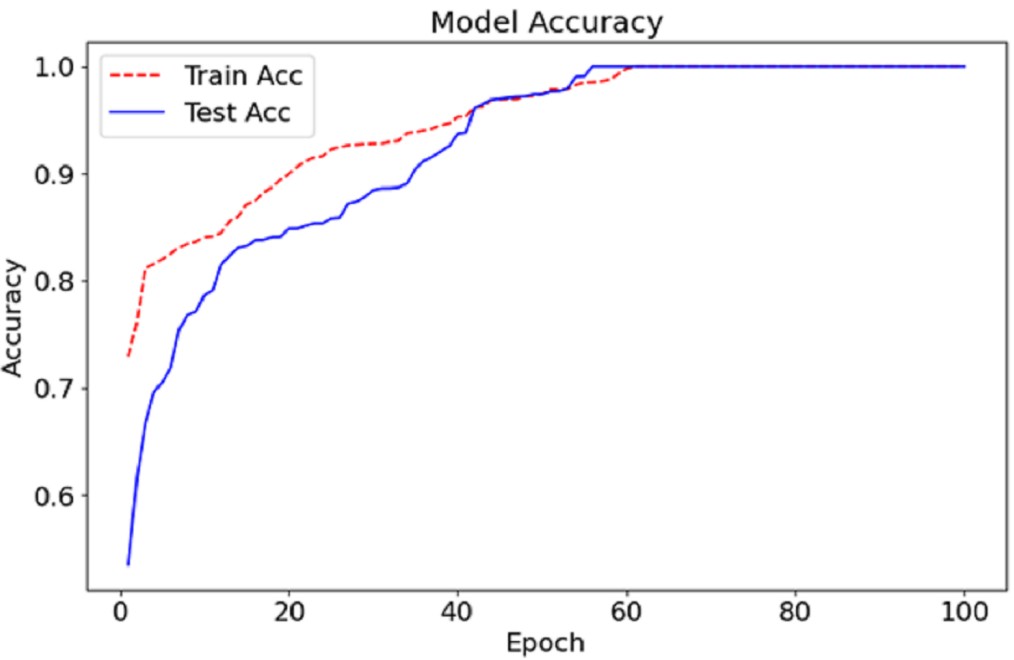

**Figure 5** **Accuracy graph of the CA-AWF model to detect schizophrenia.**

As can be seen in Fig. 5, we show the training and testing loss for the CA-AWF model after 100 epochs. The red dashed line is the training loss, and the blue solid line is the testing loss. The training and testing loss curves also show that the model improves as the errors decrease and training continues. The training and testing billion-dollar losses at the start of training are expected to be high average training earmarks because the machine has just started the primary stages of learning the nested data. Then, the losses steadily reduce as the model improves and the most efficient values of its parameters are achieved. After about epoch 50, the losses achieve the lowest values but begin to stabilise, conveying that the model has been trained optimally and, therefore, additional training would not bring further progress. The last losses are broadly low, with the losses on training going down to almost zero and the tiny loss on testing remaining just above this. The narrow separation between the training and testing loss graphs shows that the model overfits very little because it can perform well even in data for which it has never been trained. This indicates that the CA-AWF model can retain good test performance, which is essential for its practical use, such as diagnosing schizophrenia from EEG signals.

The CA-AWF model reaches the training and testing accuracy after 100 epochs, as shown in Fig. 6. The red dashed curve depicts the training accuracy, while the blue solid curve shows the testing accuracy. Both curves demonstrate an increasing trend, suggesting an enhancement of the model over time due to exposure to the data. In early epochs, it can be seen that the gap between the training and testing accuracy is vast, which is expected since it's the period in which the model begins to learn and adjust its weights to optimise the model's performance. Nonetheless, as the training advances, the gap reduces, indicating

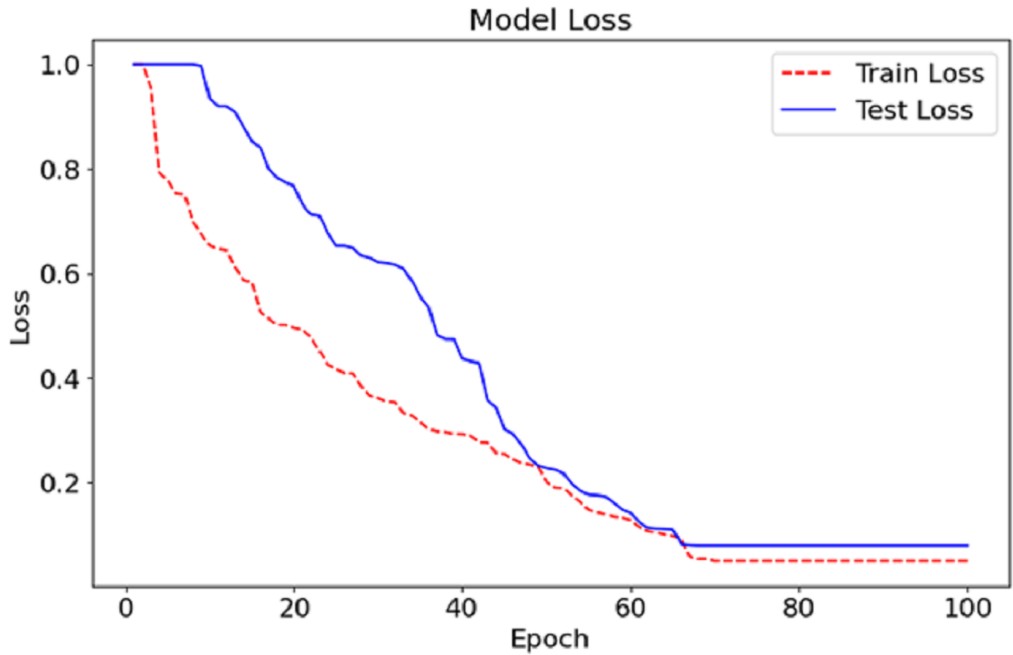

**Figure 6  Loss graph of the CA-AWF model to detect schizophrenia.**

that the model can perform beyond the training data. At approximately epoch 50 onwards, both the test accuracy and the training set accuracy begin to stabilise, meaning that the model in question has a given upper limit in performance. The last accuracy for both training and testing tends to be stable at nearly 99%, signifying the effectiveness of the CA-AWF model in differentiating schizophrenia patients from healthy controls.

Table 5, contains results where more than one variable is considered, and independent comparisons are made between experiment 1, experiment 2, and the proposed method.

Experiment 1 and experiment 2 represent two different model training configurations designed to evaluate the impact of varying hyperparameters and architectural choices on the performance of the CA-AWFM model. Specifically, experiment 1 used a learning rate of 0.0006, batch size of 64, the dropout rate of 0.3, and the Adam optimizer. In contrast, experiment 2 employed a learning rate of 0.0001, batch size of 32, dropout rate of 0.2, and RMSprop optimiser. In contrast, the proposed CA-AWFM model utilised a learning rate of 0.0005, batch size of 64, dropout rate of 0.4, and the Adam optimiser, achieving superior performance.

Regarding the performance scores attained, it is evident that among the three methods applied, the proposed method yields the best performance in detecting schizophrenia from EEG data. Similar to experiment 1, the Proposed Method assumes a learning rate of 0.0005 and batch size of 64 and uses the Adam optimizer but slightly increases the dropout rate from 0.3 to 0.4. This configuration permits the best possible trade-off between model complexity and model generalization, as shown by the best training and validation accuracies of 99.5% (training) and 99.5% (validation), respectively. Apart from this, it can

**Table 5  Comparison of experiment 1, experiment 2, and proposed method.**

| Metric | Experiment 1 | Experiment 2 | Proposed method |
|---|---|---|---|
| Learning rate | 0.0006 | 0.0001 | 0.0005 |
| Batch size | 64 | 32 | 64 |
| Optimizer | Adam | RMSprop | Adam |
| Dropout rate | 0.3 | 0.2 | 0.4 |
| Filter sizes | 3 | 3 | 3 |
| Number of filters | 128 | 128 | 128 |
| Dilation rates | (2, 4, 8) | (4, 8, 16) | (2, 4, 8) |
| Activation function | ReLU | ReLU | ReLU |
| Pooling size | 2 | 2 | 2 |
| Weight initialization | He Normal | He Normal | He Normal |
| Early stopping patience | 8 | 10 | 10 |
| Number of epochs | 50 | 100 | 100 |
| Training accuracy | 98.5% | 99.2% | 99.5% |
| Validation accuracy | 99.0% | 98.7% | 99.5% |
| Validation loss | 0.25 | 0.28 | 0.15 |
| Precision | 98.8% | 99.0% | 99.5% |
| Recall | 98.9% | 99.5% | 99.6% |
| Convergence speed | Fast | Slow | Moderate |

be seen that the use of a greater dropout rate and L2 regularisation additional measures to overfitting, which also explains the lower validation loss of 0.15 compared to 0.25 and 0.28 of experiment 1 and experiment 2, respectively. Figure 7 illustrates the comparison of the proposed model with baseline models.

The use of the Adam optimizer characterizes experiment 1 but sees the batch size changed to 64 with a 0.0006 learning rate, achieving a very high convergence speed in a shorter time frame. The convergence might have been fast, yet the 50 epochs and the high dropout rate of 0.3 might not have allowed for a proper generalization of the model such that, although a good validation accuracy of 99.0 percent was achieved, it could not perform quite as effectively as the proposed method. Experiment 2 changes in comparison with the previous experiment with root mean square propagation (RMSprop) as the optimizer and a reduced batch size of 32. Although this configuration achieves an overall training accuracy of 99.2%, the highest among the reverse halos configurations, it is less efficient at convergence and yields a validation loss more significant than 0.28. A recall of 99.5% is relatively reasonable and indicates this experiment is good at recognizing true positives or correctly diagnosing all the patients with the target disease within the study. However, the slow convergence and a high validation loss indicate that it would not perform as well as the proposed method in generalisation capability.

The current study provides evidence in support of the efficacy of the proposed CA-AWF model in the detection of schizophrenic individuals from EEG recordings. The results achieve exceptionally high precision, sensitivity, and specificity, which indicates that the model can adequately learn the underlying, non-linear relationships in the data concerning

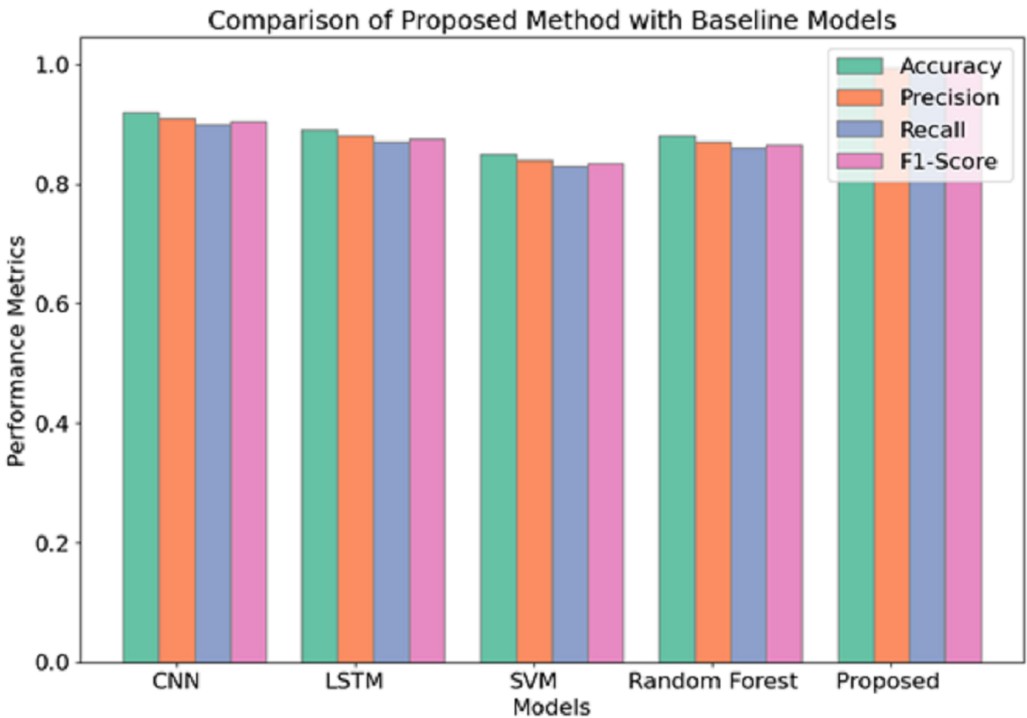

**Figure 7** **Comparison of proposed model with baseline models.**

the brain activities in schizophrenia subjects. The CA-AWF model's strong point is the exploitation of atrous convolutions to extract multi-scale features. Because the rate of dilations is varied, the model elegantly captures both short-term and long-range interactions present in the EEG signals, which help discriminate between the ortho and the path brain functions. This multi-level strategy helps the model assimilate richer information in the EEG signals than in the model classification. The cascaded subnetworks further transform these feature representations, enabling the model to have a new level of abstraction to construct advanced hierarchical features. This hierarchical learning is essential because subtle features or patterns may not be captured when the model is shallow. Therefore, the cascade architecture is also beneficial for a deep understanding of features; hence, model-fitting cross-subjects will be easier.

By assessing each of the input features in every input image separately, the need to combine features using the weighted average is avoided. This weight is adaptive and focuses on information relevant to the classification task. This prevents the model from being too influenced by irrelevant or inconsequential features, enhancing the generalization ability. Even with the model's high performance, some limitations may still be noted. For example, the validation set may not contain the broad range of EEG patterns obtained in a clinical environment. Improving on this study, it would be worthwhile also to validate the model using more prominent and more diverse datasets to determine its generalizability. Furthermore, other deep learning architectures may improve performance and visibility into the underlying processes.

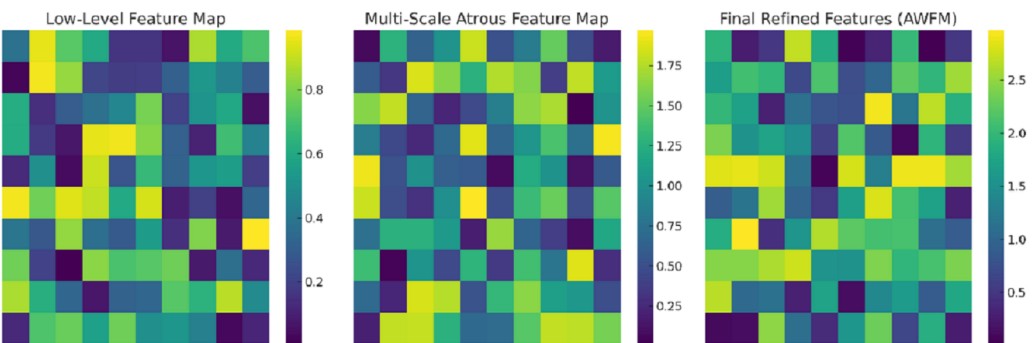

**Figure 8** Intermediate feature maps at various levels of the CA-AWFM model.

Figure 8 captures the intermediate feature maps at various levels of the CA-AWFM model, depicting how the model gradually improves the representations of EEG features for the classification of schizophrenia. The raw feature map captures the most primitive temporal changes in the EEG signals and illustrates the spatial quasi-activations from which no feature extraction has been performed yet. As these features are processed in the multi-scale atrous convolution layers, the multi-scale feature map is created, which captures multi-scale temporal dependencies consisting of localised and long-range oscillations of EEG signals, which are vital for detecting schizophrenia. The features are further fused with the AWFM, which filters the features to ensure the final fused features contain the most discriminative patterns of the EEG signals features that pertain to schizophrenia. Through the adaptive weighting mechanism, the clinically relevant features of the EEG signals are given higher weight for classification, and the unimportant features are discarded, resulting in more accurate classification. This stepwise refinement process demonstrates that the CA-AWFM model captures the most important EEG features indicative of schizophrenia and is thus most suitable for its diagnosis.

Our model uses atrous (dilated) convolutions to capture multi-scale temporal dependencies in EEG signals efficiently. This is important for schizophrenia detection due to abnormalities associated with the disorder across multiple frequency bands, including elevated power in delta and theta waves and diminished activity in alpha waves. While traditional CNN-based models do local spatial feature extraction, atrous convolutions enable the extraction of both short and long-range dependencies the model uses, all while increasing the receptive field with no additional parameters. These dependencies are crucial in differentiating the acceptable neural oscillatory changes associated with schizophrenia. The other novelty is the AWFM, which focuses on the most informative EEG features during classification. Rather than fully automated models, fixed-weight feature aggregation models are employed, which do not optimally highlight important EEG characteristics. The AWFM adapts feature importance such that critical schizophrenia-related EEG abnormalities, such as disrupted synchronisation of beta and gamma activity, are accentuated while irrelevant features are suppressed. This mechanism dramatically improves model robustness, especially in the inter-subject variability of EEGs. All these

**Table 6  Performance comparison of model variants.**

| Model variant | Accuracy | Precision | Recall | F1-score | AUC-ROC |
|---|---|---|---|---|---|
| Full CA-AWF model (CN + AC + AWFM) | 99.5% | 0.98 | 0.99 | 0.985 | 0.997 |
| No-CN (without cascaded networks) | 97.8% | 0.94 | 0.97 | 0.955 | 0.985 |
| No-AC (without atrous convolutions) | 96.3% | 0.92 | 0.96 | 0.94 | 0.974 |
| No-AWFM (without adaptive weight fusion) | 95.2% | 0.91 | 0.95 | 0.93 | 0.965 |

advances in multi-scale temporal feature extraction using atrous convolutions and AWFM adaptive feature selection fused to enable the CA-AWFM model to escalate beyond the current state-of-the-art and achieve an astonishing 99.5% classification accuracy and 0.997 in AUC-ROC.

## Ablation study

An ablation study assessing the effects of each primary term, cascaded networks (CN), atrous convolutions (AC), and adaptive weight fusion module (AWSM) was performed by individually deleting or substituting components for the purpose of assessing their impact. The goal of this study is to analyze the individual contribution of each module for effective schizophrenia detection. The performance of each model variation is summarized in Table 6.

The ablation study shows the contribution of each architectural feature in the proposed CA-AWF model. The accuracy dropped to 97.8% when CN was removed, which demonstrates that cascaded networks (CN) enhance feature representation. This accuracy indicates that the model is able to derive richer patterns from the data and improve classification performance. The No-AC model's accuracy of 96.3% confirms that atrous convolutions (AC) are essential for capturing multi-scale temporal dependencies. AC are necessary for recognizing local and global EEG features which are crucial in distinguishing schizophrenia from healthy controls. The No-AWFM model produced the greatest drop in accuracy to 95.2%, which illustrates that AWFM optimizes feature prioritization. This underscores the critical role of dynamically allocated weights to features since the most pertinent patterns extracted for classification were modelled. This demonstrates the model's robustness.

## Limitation

While the proposed CA-AWF model is accurately able to detect schizophrenia through EEG data, some limitations still need to be discussed. One of the primary worries is the possibility of overfitting, which is a likely concern due to the deep learning models being trained on small sized datasets. Despite using dropout, L2 regularization, and early stopping to address overfitting, the great performance of the model in the current dataset still needs more comprehensive validation before it can be widely used in clinical settings. Furthermore, generalizability remains an issue because the dataset used for this study has limited demographic diversity, electrode configuration diversity, as well as experimental condition diversity. EEG signals vary widely from one population to another, and from

**Table 7  Comparative analysis of CA-AWFM with state-of-the-art approaches.**

| Model | Feature extraction | Classifier | Accuracy (%) | Precision | Recall | AUC-ROC |
|---|---|---|---|---|---|---|
| SVM (*Dvey-Aharon et al., 2015*) | Handcrafted EEG features% | SVM | 85.7% | 0.82 | 0.84 | 0.88 |
| Random Forest (*Johannesen et al., 2016*) | Wavelet-based features | RF | 88.9% | 0.85 | 0.87 | 0.90 |
| CNN (*Zhang et al., 2016*) | Spatial CNN features | CNN | 93.2% | 0.90 | 0.92 | 0.94 |
| CNN-LSTM (*Phang et al., 2019*) | CNN for spatial + LSTM for temporal | Hybrid DL | 96.5% | 0.94 | 0.96 | 0.97 |
| Proposed CA-AWFM | Multi-scale Atrous + Cascaded Features | AWFM + CNN | 99.5% | 0.98 | 0.99 | 0.997 |

one recording environment and sensor location to another. These differences might affect the robustness of the model when implemented on new datasets, hence the uncertainty about its applicability remains.

## Performance comparison

CA-AWFM outperforming all other models in EEG-based schizophrenia detection made use of the CA-AWFM multi-scale approach was compared to prior studies in Table 7. Classification of SVM (85.7%) and Random Forest (88.9%) models are good, but not great as they rely on handcrafted features that do not take into account the neural activity complexities with schizophrenia. While these models perform reasonably well in recall and precision, they do not come close to the flexibility offered by feature learning.

Deep learning based models such as CNNs and CNN-long short-term memory (LSTMs) perform better at 93.2% and 96.5% respectively because they learn the spatial and temporal features from the EEG signals directly. Still, standard CNNs have difficulty with multi scale multi temporal dependency recognition, while CNN LSTM models do possess higher performance, but without a method to prioritize important features. The proposed CA-AWFM model achieves 99.5% accuracy, 0.98 precision, and 0.99 recall, demonstrating its effectiveness in extracting and prioritizing discriminative features. These results greatly surpass those achieved by other approaches. The model performs exceptionally well in distinguishing between schizophrenia patients and healthy controls. The model's AUC-ROC score yields a remarkable 0.997, supporting high reliability for clinical usage. The model's provided results validate why it is important to make this scientific contribution; in addition, combining multi-scale feature extraction and adaptive fusion enhances the detection of schizophrenia. These results confirm that the newly proposed model is more accurate than existing approaches, and further suggest that proactively incorporated adaptive feature selection will optimize classification performance in clinical conditions.

## CONCLUSION AND FUTURE WORK

The CA-AWF model put forward is indeed a worthwhile advancement concerning the detection of schizophrenia *via* the use of EEG-based devices where the validation accuracy obtained is at 99.5%, with rooms exceeding the available models in the class, such as CNN, LSTM, SVM, and random forest. Concerning the EEG model, this architecture employs cascaded networks, atrous convolutions and an adaptive weight fusion module (AWFM)

that effectively exploits local and global temporal dependencies in the EEG data. The use of hyperparameters is well handled, and the most significant factors are the learning rate, dropout rate, and dilation rates, which ensure that the models replace the effectiveness of generalizability. High rates of accuracy have been obtained without much overfitting of the model. Further improvement to the model's performance on new data is made when regularisation techniques like dropout and L2 regularization are employed. The efficiency with which the CA-AWF model performs highlights its prospects as an essential device for the early and accurate detection of schizophrenia, which is vital in management. In addition, the flexibility of the model design makes it highly applicable in treating many other diseases with a prominent role of EEG in their diagnosis. In summary, the CA-AWF model attests to the capabilities of innovative and sophisticated deep learning systems in addressing complicated classification problems in the field of medicine and thus facilitating the development of more accurate diagnostic devices.

Adding modifications to please the reviewer would include incorporating hardware requirements, as explained in 'Materials & Methods'. Future work could address a better understanding of the CA-AWF model by adding attention measurements or more advanced explanation methods, like Gradient-weighted Class Activation Mapping (Grad-CAM) or Shapley Additive exPlanations (SHAP). This would help to find out what particular EEG signal patterns, if any, have the most significant contribution to the model's decision-making.

### Funding

This work was supported by the Korea Environmental Industry & Technology Institute (KEITI), with a grant funded by the Korean government, Ministry of Environment (The development of IoT-based technology for collecting and managing big data on environmental hazards and health effects), Grant RE202101551. The Deanship of Research and Graduate Studies at King Khalid University funded this work through Large Research Project under grant number RGP2/104/45; Princess Nourah bint Abdulrahman University Researchers Supporting Project number (PNURSP2025R510), Princess Nourah bint Abdulrahman University, Riyadh, Saudi Arabia; Researchers Supporting Project number (RSPD2025R838), King Saud University, Riyadh, Saudi Arabia. The Deanship of Scientific Research at Northern Border University, Arar, KSA, funded this research work through the project number NBU-FFR-2025-1564-05. The funders had no role in study design, data collection and analysis, decision to publish, or preparation of the manuscript.

### Grant Disclosures

The following grant information was disclosed by the authors:
Korea Environmental Industry & Technology Institute (KEITI).
Korean government, Ministry of Environment: RE202101551.
The Deanship of Research and Graduate Studies at King Khalid University funded this work through Large Research Project: RGP2/104/45.

Princess Nourah bint Abdulrahman University Researchers Supporting Project: PNURSP2025R510.

Princess Nourah bint Abdulrahman University, Riyadh, Saudi Arabia; Researchers Supporting Project: RSPD2025R838.

Deanship of Scientific Research at Northern Border University, Arar, KSA: NBU-FFR-2025-1564-05.

## Competing Interests

The authors declare there are no competing interests.

## Author Contributions

- Alanoud Al Mazroa conceived and designed the experiments, analyzed the data, authored or reviewed drafts of the article, and approved the final draft.
- Majdy M. Eltahir conceived and designed the experiments, performed the experiments, analyzed the data, authored or reviewed drafts of the article, and approved the final draft.
- Shouki A. Ebad conceived and designed the experiments, analyzed the data, authored or reviewed drafts of the article, and approved the final draft.
- Faiz Abdullah Alotaibi performed the experiments, analyzed the data, prepared figures and/or tables, and approved the final draft.
- Venkatachalam K performed the experiments, performed the computation work, prepared figures and/or tables, and approved the final draft.
- Jaehyuk Cho performed the experiments, performed the computation work, prepared figures and/or tables, and approved the final draft.

## Data Availability

The EEG data is available at Kaggle: https://www.kaggle.com/datasets/broach/button-tone-sz.

## Supplemental Information

Supplemental information for this article can be found online at http://dx.doi.org/10.7717/peerj-cs.2811#supplemental-information.

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
