# Peer review of "EEG-based schizophrenia diagnosis using deep learning with multi-scale and adaptive feature selection"

_PeerJ Computer Science, doi:10.7717/peerj-cs.2811_

## Round 0.1 · original submission · Major Revisions

Dear authors,

Thank you for submitting your article. Reviewers have now commented on your article and suggest major revisions. We do encourage you to address the concerns and criticisms of the reviewers with respect to reporting, experimental design and validity of the findings and resubmit your article once you have updated it accordingly. In addition, it is imperative that the manuscript undergoes thorough proofreading. It is evident that the article contains a multitude of spelling and grammatical errors. It is important to pay attention to the use of the space character. Explanation of the equations should be checked. All variables should be written in italics as in the equations. Their definitions and boundaries should be defined. Necessary references should also be given. Many of the equations are part of the related sentences. Attention is needed for correct sentence formation.

Warm regards,

Reviewer 1 ·

Basic reporting

I have reviewed the manuscript titled "EG-Based Schizophrenia Detection Using Cascaded Networks with Atrous Convolutions and Adaptive Weight Fusion." The study presents a novel deep-learning model (CA-AWFM) for classifying schizophrenia from EEG signals, leveraging cascaded networks, atrous convolutions, and an adaptive weight fusion module. The authors report achieving 99.5% accuracy, which is an impressive result for this domain. However, the manuscript requires substantial revisions to improve clarity, reproducibility, and alignment with research standards.

Major revisions are required. Below are detailed suggestions to address the manuscript's limitations:

Include an ablation study to assess the contribution of each architectural component (cascaded networks, atrous convolutions, and the adaptive weight fusion module) to the model's performance.
Clearly define the limitations of the study, including potential overfitting risks and generalizability concerns, and suggest future research directions for validation on larger and more diverse datasets.
Ensure consistent use and definition of abbreviations throughout the manuscript, such as "EEG" and "CA-AWFM."
Add a block diagram or pseudo-code to illustrate the model’s architecture and data flow for improved understanding.
Expand the introduction to better articulate the research gap, providing a more comprehensive overview of existing methods and how the proposed method advances the field.
Provide a direct link or access to the EEG dataset used and include the codebase for reproducibility.
Improve the method section with additional details, particularly on hyperparameter tuning and preprocessing steps. Include comparisons in the discussion section with existing state-of-the-art approaches.
Expand the experimental results section to include further analysis of parameter optimization strategies and their impact on the model's performance.
Clarify the novelty of the proposed method compared to existing approaches, focusing on how the inclusion of atrous convolutions and adaptive weight fusion differentiates it.
Explain how the features extracted by the model align with known EEG patterns in schizophrenia. Consider visualizing intermediate feature representations to support interpretability.
Provide a detailed commentary on tables and figures, ensuring all results are thoroughly interpreted and contextualized within the study's goals.
Revise the manuscript title to improve clarity and avoid ambiguity. The term "EG" is not defined or widely recognized, which may confuse readers.
Additionally, I recommend the authors review the following related work to enhance the literature discussion:Epilepsy detection in 121 patient populations using hypercube pattern from EEG signals .

Experimental design

As above

Validity of the findings

As above

Reviewer 2 ·

Basic reporting

This paper has some merits, but significant changes has to be done.

1. Proof reading is required. There are multiple typos. There is a mistake in title. AWFM is defined multiple times. Authors should follow some standard style of using Acronyms.
2. Introduction needs to be improved. Recent works form 2024 and 2023 needs to incorporated. Literature review is outdated.
3. Description of the dataset needs to be expanded. More details such as dataset size, number of channels, sampling rate and other important details needs to be added.
4. State of the art comparison needs to be improved.

Experimental design

Authors should explain the importance of AWFM experimentally. How AWFM contribute to the model.
Try to include ablation study.

Validity of the findings

It is not clear for me how the experiments are performed.
What are experiments 1, and 2 in Table 4? There is no comparison in table4. But authors named it as comparison.
How did the authors partitioned the dataset?
Try to include ablation study.

Additional comments

Some major rewriting is required.

---

## Round 0.2 · accepted · Accept

The paper was very well improved, so can be accepted!

Reviewer 1 ·

Basic reporting

The authors have completely addressed all my comments, and I have no further concerns. Therefore, I recommend accepting the paper.

Experimental design

The authors have completely addressed all my comments, and I have no further concerns. Therefore, I recommend accepting the paper.

Validity of the findings

The authors have completely addressed all my comments, and I have no further concerns. Therefore, I recommend accepting the paper.